# Non-Occupational Exposure to Pesticides: Experimental Approaches and Analytical Techniques (from 2019)

**DOI:** 10.3390/molecules26123688

**Published:** 2021-06-16

**Authors:** Lucía Vera-Herrera, Daniele Sadutto, Yolanda Picó

**Affiliations:** Food and Environmental Safety Research Group of the University of Valencia (SAMA-UV), Desertification Research Centre (CIDE), CSIC-GV-UV, Moncada-Naquera Road km 4.5, Moncada, 46113 Valencia, Spain; vehelu@uv.es (L.V.-H.); sadutto@uv.es (D.S.)

**Keywords:** pesticide residues, human health, direct estimation, indirect estimation, estimated intakes, environmental exposure, wastewater-based epidemiology

## Abstract

Background: Pesticide residues are a threat to the health of the global population, not only to farmers, applicators, and other pesticide professionals. Humans are exposed through various routes such as food, skin, and inhalation. This study summarizes the different methods to assess and/or estimate human exposure to pesticide residues of the global population. Methods: A systematic search was carried out on Scopus and web of science databases of studies on human exposure to pesticide residues since 2019. Results: The methods to estimate human health risk can be categorized as direct (determining the exposure through specific biomarkers in human matrices) or indirect (determining the levels in the environment and food and estimating the occurrence). The role that analytical techniques play was analyzed. In both cases, the application of generic solvent extraction and solid-phase extraction (SPE) clean-up, followed by liquid or gas chromatography coupled to mass spectrometry, is decisive. Advances within the analytical techniques have played an unquestionable role. Conclusions: All these studies have contributed to an important advance in the knowledge of analytical techniques for the detection of pesticide levels and the subsequent assessment of nonoccupational human exposure.

## 1. Introduction

Pesticides are particularly important among agrochemicals as they are widely used in modern agriculture to control weeds and different pests affecting crops [1,2]. They play an important role in improving agricultural production providing important benefits for humanity. According to the Food and Agriculture Organization (FAO), a “pesticide means any substance, or mixture of substances, of chemical or biological ingredients intended for repelling, destroying, or controlling any pest, or regulating plant growth” [3]. 

Pesticides differ in their physical and chemical properties, which define their mechanisms of action on target organisms [4,5]. A distinction is made between natural (plant- or mineral oil-based) and synthetic pesticides [6]. The latter are classified into many categories depending on their chemical composition. The four most well known are organochlorine (OCPs), organophosphate (OPPs), carbamates, and pyrethroid pesticides [7]. However, today, there are many more, such as triazines, thiocarbamates, pyrazoles, coumarin derivatives, ureas, and strobilurins [8], that form a mishmash of chemical structures whose only common thread is their ability to eliminate pests. Many times, to simplify, pesticides are distinguished according to the organism they kill: insecticides, herbicides, fungicides, rodenticides, nematicides, etc. [5].

According to the FAOSTAT database, global pesticide use increased in the 2010s by 50% with respect to the 1990s, with pesticide use per area of cropland increasing from 1.80 to 2.66 kg/ha. In contrast, pesticide use has remained stable in recent years, due to a slight decrease in herbicide use (from 1.25 Mt to 1.22 Mt in 2018 compared to 2017) [9]. However, the application and release of these pollutants into the environment continue to occur in large quantities. In fact, it is estimated that only 1% of the active ingredient acts on the target crop and 99% ends up in the environment [10]. Furthermore, data have indicated that farmers and other producers apply 40 billion USD worth of pesticides per year, of which only 2% are recently developed biopesticides with fewer contaminants [11]. 

Despite improved crop yields, the introduction of these compounds causes serious hazards to the environment. Pesticides can contaminate water bodies, soils, sediments, and biota, including fauna and flora [12]. The main sources of pesticide discharge are agricultural fields, atmospheric precipitation (including accumulation in dust and aerosols), and untreated sewage from industrial and urban centers and hazardous-waste-disposal sites [13]. Their frequent use, together with their persistence in natural matrices and the capacity of some of them for biomagnification and bioaccumulation [14,15], makes them ubiquitous, highly contaminant, and dangerous for the environment. 

Pesticides also have adverse effects on human health. Cases of acute pesticide poisoning (APP) account for significant mortality globally, especially in developing countries [16], affecting mainly agricultural workers and populations located in cultivated environments [17]. Numerous diseases have been linked to pesticide exposure. In particular, direct exposure to these pollutants is recognized as the main cause of cancer worldwide, as well as being linked to other diseases such as respiratory and neurological disorders, diabetes, reproductive syndromes, and oxidative stress [18]. For this reason, pesticides are considered as dangerous substances by Directive 2006/11/EC [19]. Several legislations and directives have been established to (i) regulate the placing on the EU market of fertilizers (Regulation 2019/1009/EU) and biocidal products (Regulation 528/2012/EU), (ii) set maximum residue levels of pesticides in food and feed of animal and plant origin (Regulation 396/2005/EC), and (iii) establish a framework for community action to achieve the sustainable use of pesticides (Directive 2009/128/EC) [20,21,22,23]. In addition, the European Commission publishes statistics on pesticide sales in Europe, analyzing geographical location, year, and pesticide groups, as well as periodically updates the “pesticide residues database”, classifying these chemicals as “approved” or “nonapproved” and establishing their MRLs in any type of food [24]. 

In this context, the human health risk is directly related to the level of pesticide exposure of the individuals. In the first instance, occupational exposure, mainly direct exposure [25], concerns principally workers in the agricultural and chemical sector [26]. This exposure occurs mainly through inhalation of residues from aerial emissions produced during spray application, and through dermal exposure during any contact with pesticides (loading and cleaning equipment, deposition of particles during work, etc.) [25,27]. Hygiene and safety measures, as well as the use of personal protection equipment (PPE), are essential to reduce exposure in these circumstances [27]. This occupational exposure is considered outside the scope of this study. On the other hand, the number of studies assessing nonoccupational exposure has increased in recent years. Indirect exposure through food intake, as well as accidental ingestion/inhalation of contaminated water, soil, and sediments, is also considered as an important potential routs for pesticide exposure at an individual level [28]. Interviews and questionnaires to estimate food consumption and assumption of a worst-case scenario in which all commodities were at the maximum residue limit (MRL) are widely used as assessment techniques [29]. However, these techniques have certain limitations and weaknesses [29,30], since they are restricted to specific and short timeframes and do not take into account the exact frequency of pesticide use in their environment or the real distance to the crops, among other factors. Thus, an improved understanding of the population’s environmental exposure to pesticides can be reached determining the actual concentrations of pesticide residues in food, environment, or biological sample. The possibility to determine pesticide residues is very related to the advances in analytical techniques. Interestingly, Dereumeaux et al. [29] compiled several epidemiological studies demonstrating how the population living close to agricultural lands is more exposed to pesticides, in terms of high pollutant levels in environmental and biological matrices, using a combination of measured concentration, geographical information systems (GIS), and modeling. This is one of the several studies that pointed out the importance of quantifying pesticide residues in different environmental compartments, as well as in biological fluids, for assessing health risks and for developing appropriate prevention and mitigation strategies. 

Therefore, the main aim of this review was to provide a global overview using the most recent literature of the role that these analytical improvements within pesticide residue determination has played in assessing nonoccupational human exposure to pesticide, as well as the status of techniques used for this assessment, so as not to perform an exhaustive review of the whole literature. The different approaches for such studies, the main matrices, and the commonly used methodologies, including pesticide extraction and subsequent determination, were considered. The different integrated mathematical approaches to assess the exposure to these pollutants through the main pathways were critically analyzed. This review performs an examination of studies published from January 2019 to March 2021, even though a few others have also been included because of their relevance to the direct and indirect evaluation of exposure to pesticides.

## 2. Materials and Methods

The search was conducted on the database Scopus (Elsevier), with the following input: “pesticide environmental exposure assessment” [title/abstract; all fields], covering articles published between January 2019 and March 2021. In addition, two additional searches were carried out; the first aimed at collecting reviews published in the same period related to our scope of study (“pesticide exposure” and “pesticide detection extraction”; [title/abstract; review]), and the second aimed at collecting all available wastewater-based epidemiology studies published between January 2018 and March 2021 (“pesticide exposure wastewater”; [title/abstract; all fields]) since no articles assessing human exposure to pesticides using this approach were found in the last 2 years. More than 1100 works were reviewed. Meta-analysis articles were not included in this review. The selection criteria to choose the studies were based on (i) the assessment of exposure (nonimpact or adverse health effects) to pesticides, always on a human population level, excluding environmental studies assessing the exposure in biota, (ii) the main environmental matrices (water, soil, sediment, and air), in addition to others considered relevant such as dust, food, and drink, and (iii) the analysis of pesticide biomarkers in human biological matrices and in the inputs of sewage treatment plants. In addition, some works outside the chosen period, considered relevant to our study, were also cited.

## 3. Environmental and Dietary Exposure to Pesticides

According to the search criteria described above, a total of 73 articles regarding the analysis of nonoccupational exposure to pesticides were included in this review. These articles were categorized according to the approach used to assess population exposure. Human exposure to pesticides can be estimated through direct and indirect approaches. Figure 1 schematizes the main assessment models for pesticide exposure, together with the main pathways. 

Indirect approaches, also called “external exposure approaches” [28], estimate the exposure of a population [31] through the measurements of pesticide residue levels in food and the environment. Such assessments are based on the interactions of the human being with the environment and determine the amount of pesticides contacted and the duration of the contact [32]. Environmental sampling involves water resources (mainly tap water, groundwater, rivers, and lakes), soils, sediments, and air particles. Recent studies have also included indoor and road dust [33,34]. In addition, as mentioned above, indirect assessment models also include food, as fruits, vegetables, and cereals treated with pesticides have also been proven to be significant sources of pesticides. Furthermore, the pesticide residues in abiotic environmental samples also bioaccumulate in biota. Dietary intake of animal products, including meat and subproducts from farm animals, fish, and seafood, is considered an important route of exposure for population [35]. Although pesticide residues are in low concentrations in food, the risk to human health is high due to their consumption over a lifetime [28]. 

On the other hand, direct approaches, also called “internal dose approaches” [28], assess the nonoccupational exposure via measurements of specific human biomarkers in biological matrices. These assessment models are considered the best sources of data for estimating actual individual exposure. Pesticide biomarkers can be the unaltered parent compound or the metabolites, which have a different concentration depending on the external exposure [31]. Biomonitoring is mainly carried out in urine samples, as well as in other biological matrices such as blood, plasma, serum, hair, breast milk, and placenta [36]. 

Determining pesticides analytically requires knowledge on the physicochemical characteristics of the target compounds and the composition of the studied matrix. The sample preparation, the first step in the determination of pesticides and metabolites, is necessary for enriching and purifying the analytes [7]. The principal methods of extraction are solid–liquid extraction (SLE), solid-phase extraction (SPE), solid-phase microextraction (SPME), and dispersive liquid–liquid micro-extraction (DLLME). The most popular method based on SLE is the Quick, Easy, Cheap, Effective, Rugged, and Safe (QuEChERS) method, which involves SLE followed by a clean-up process, usually using dispersive solid-phase extraction (d-SPE) [1]. The next steps of the analytical process are the separation and detection of pesticide residues. In recent years, the most used strategies for separating pesticide residues in a prepared sample have been gas chromatography (GC) and liquid chromatography (LC) due to their versatility and separation abilities coupled to mass spectrometry detectors (high and low resolution and in tandem) [5]. Other detectors also applied include electron capture detection (ECD), fluorescence programmable detection (FPD), and dual flame photometric detection (DFPD) [12]. The selected analytical methods require an accurate validation that is based on the use of analytical standards and involves the establishment of linearity, matrix effects, accuracy (commonly as recoveries) [12], sensitivity (limits of detection and quantification of the target analytes), and precision (by assessment of repeatability and reproducibility) [31].

The main mathematical approach to assess pesticide exposure through food is based on calculating the estimated daily intake (EDI) and the hazard index (HI) [28]. Long-term hazards are evaluated using the acceptable daily intake (ADI), a reference point established by the Joint FAO/WHO Meeting on Pesticide Residues (JMPR) that specifies the maximum permitted daily intake for a person over a lifetime without major risk to the individual [37]. The maximum residue limit (MRL) means the maximum concentration of a pesticide residue (mg/kg) legally permitted in food and animal feeds. In this context, pesticide residue includes any derivatives of a pesticide, such as conversion products, metabolites, reaction products, and impurities considered to be of toxicological or ecotoxicological significance [38]. MRLs are established in each country or association of countries (such as the European Union) to ensure legal compliance, although most of them are based on those recommended by the Codex Alimentarius Commission (CAC) [1,37]. CAC MRLs are based on toxicological and agronomic criteria and can be used to calculate the worst-case scenario of exposure. In indirect methods, the pesticide levels determined in food are used to calculate the estimated daily intake (EDI), according to Equation (1) [28].
EDI (µg/kg bw/day) = ∑ [RLi (µg/kg) × Fi (kg/day)]/BW (kg bw),(1)
where RLi is the pesticide residue level, Fi is the consumption rate of food, and BW is the mean body weight of the study population. Acute dietary risk (RQa) has also been evaluated in some studies through the calculation of the national estimated short-time intake (NESTI), which requires the highest residue value (HR) and the large portion of food for general consumer (LP). 

The cumulative human exposure to pesticides detected in surface water, groundwater, and drinking water can be determined using different mathematical approximations. These include estimating the chronic daily intake (CDI) or the estimated daily intake (EDI) using the reference dose (RfD) or the ADI for each compound and the concentration in each water sample. Some studies have considered both ingestion and dermal contact and calculated the noncarcinogenic health risk (HI) and the cancer risk (CR) associated with drinking water and bathing exposure. In sediment and soil, human exposure was assessed covering the cancer and noncancer risk, also called total lifetime carcinogenic risk (TCLR) and total noncarcinogenic hazard quotient (THQ) [39,40,41]. Thus, the human exposure and health risk were assessed as a sum of the risk for inhalation, ingestion, and dermal exposure.

The most common approach to estimate the inhalation exposure of atmospheric pesticides is based on the calculation of the inhalation daily intake doses (DIinh), which requires the mean concentration of the analytes in inhaled air, as well as the inhalation rate and the mean body weight. Some studies have also considered the exposure duration and its frequency. The human health risk is assessed using the hazard quotient (HQ), which results from dividing the DIinh by the health base reference values (HBRV).

For the calculation of EDI in direct methods through biomonitoring studies, the levels of metabolites measured in urine are normally converted into daily intake of parent compound using Equation (2) [28].
EDI (µg/kg bw/day) = [C_U_ (µmol/L) × V_U_ (L) × MW_P_ (g/mol)]/F_UE_ × BW (kg),(2)
where C_U_ is the molar concentration of the nonspecific and/or specific pesticide metabolites, V_U_ is the total volume of urine excreted within 24 h, MW_P_ is the molecular weight of the parent compound, and F_UE_ is the urinary excretion factor of the parent compound. In both approaches, the risk quotient (HQ) is calculated by the ratio between EDI and ADI [42]. 

The main approaches to assess the health risk of pesticides in dust are related to dermal and ingestion exposure pathways. Yadav et al. [43,44] estimated the dust ingestion and the dermal absorption risk using USEPA’s risk assessment guideline. Anh et al. [34] assessed the daily intake doses of pollutants via road dust ingestion also taking into account the fraction of time the individual spent outdoors. 

In addition to mathematical approximations, many studies have assessed the exposure of a population to pesticides using questionnaires and personal interviews. These tools provide information about diet, occupation, education, residence, nearby agricultural areas, pesticide use, exposure duration, etc. Figure 2 shows the main steps necessary to conduct a comprehensive pesticide exposure assessment of a population. 

Furthermore, in WBE, for the back-calculation of population exposure and intake, the four referenced articles followed the same methodology [45,46,47,48]. First, they calculated the daily mass load (MLday) of the selected biomarkers, following Equation (3).
MLday (mg/day) = Conc (mg/L) × V (L/day),(3)
where Conc is the total concentration of the target analyte in influent wastewater, and V is the volume of wastewater received by the WWTP per day. 

Then, the human intake (Q) was calculated following Equation (4).
Q (mg/day/1000 inh) = MLday/P × CF × 1000,(4)
where mass loads calculated were normalized to the number of people served each day by the WWTP (P), and specific correction factors (CF) were applying according to the percentage of excretion of each compound in human urine.

In this review, the 73 selected articles include 37 environmental monitoring studies (19 on food, 11 on water, soil, and sediment samples, four on air, three on dust, and one on silicone wristbands), 30 biomonitoring studies, two studies assessing urine and air levels together, and four wastewater-based epidemiology (WBE) studies. 

## 4. Indirect Approaches: Environmental Monitoring

### 4.1. Analysis of Pesticide Levels in Water, Sediment, and Soil

A total of four studies assessing pesticide levels in water, three in soil, one in sediment, and three in both water and sediment samples were included. Three of these studies involved pesticide multi-analyses: Huang et al. [49] analyzed 56 pesticides in groundwater samples, covering OCPs, OPPs, and carbamates. Dong et al. [50] evaluated the levels of 65 pesticides in surface water and sediment samples, including OCPs, OPPs, triazines, and amides. Lastly, Bradley et al. [51] selected 224 and 119 target pesticides in water surface and sediment matrices, respectively, covering a wide spectrum of compounds. The remaining studies screened specific groups of compounds, such as neonicotinoid insecticides (NEOs) (e.g., acetamiprid, clothianidin, and imidacloprid) [52,53], OCPs and some specific metabolites [39,40,41,54,55], and OPPs and OCPs, among others [56].

In relation to the extraction strategies, six different techniques were identified. Figure 3 shows the percentage of studies depending on the matrix and the extraction method. Pesticides analyzed in water were extracted by LLE or SPE. In LLE, reported solvents were dichloromethane or a mixture of ethyl acetate/methylene. In addition, Lu et al. [52] and Jin et al. [54] proposed cleaning up the organic phase obtained through a silica gel column. In SPE, the water samples were passed through HLB or C18 sorbents, and then eluted with acetonitrile, methanol, dichloromethane, or ethyl acetate solutions. On the other hand, pesticides analyzed in sediment and soil were commonly extracted by pressurized liquid extraction (PLE), QuEChERS, SPE, and Soxhlet. Reported PLE methodologies were similar, using the accelerated solvent extractor to perform the extraction process at high pressure and temperature (over 1500 psi and 100–120 °C). Then, the extracts were concentrated to near dryness, reconstituted in different solvents, and purified by SPE to reduce matrix interferences using two different cartridges (Florisil^®^ and ENVI-CARB/PSA). Tran et al. [40] also passed the extract through an activated copper column (Cu powder activated with a 20% HCl solution) prior to the SPE to remove sulfurs. QuEChERS extraction was carried out using 1% acetic acid in acetonitrile as solvent sodium acetate for buffering at pH 4.5 and sodium chloride and magnesium sulfate for salting out, followed by dSPE. This is the acetate-buffered version of the QuEChERS recommended by the American Official Association of Analytical Chemists (AOAC) [57]. Lastly, Ali et al. [41] applied a solid–liquid extraction (SLE) method using continuous Soxhlet extraction with acetone/*n*-hexane (1:1, *v*/*v*) solvent. Sediments are quite complex matrices that strongly retain pesticides in the organic matter (humic and fulvic acids) or in the silts, and they sometimes require the application of exhaustive extraction to properly recover pesticides.

Although several studies have determined LC–MS to be more sensitive than GC–MS for detecting the main classes of currently used pesticides, only three studies exclusively used LC–MS to analyze extracted pesticides, where the MS/MS was conducted with an electrospray ionization (ESI) source in the positive ion mode (Bradley et al. [51] also worked in the negative ion mode), with multiple selected reaction monitoring (MRM). In the other studies, the separation and the detection of the analytes were performed by GC–MS. Lastly, Bhandari et al. [41] applied both GC–MS for volatile pesticides and LC–MS/MS for more polar ones. Information regarding the analytical procedures, as well as the obtained recoveries and detection limits, is summarized in Table 1. Additional information (separation columns and mobile phases) is shown in Appendix A). Some studies did not include quality assurance and quality control (QA/QC) tests.

### 4.2. Analysis of Pesticide Residues in Atmospheric Particulate Matter

During the pesticide application in crops by aircraft or land spraying, 30% to 50% of the applied amount can remain in the atmosphere [58]. Accordingly, in the last few years, the study of the atmospheric levels of pesticides has received more attention. This review includes six studies assessing pesticide levels in atmospheric particulate matter, including two studies combining the measures of ambient air and urinary levels. The publications covered mainly OPPs and OCPs, although two studies focused on the detection of some relevant NEOs such as imidacloprid, acetamiprid, clothianidin, and thiamethoxam [59,60]. Air samples were collected by suspended particulate samplers (high or low volume, according to the study) equipped with quartz or glass fiber filters (GFFs) and polyurethane foam (PUF) cartridges. Two sampling methods were identified: the active air sampling method, which requires the use of a pumping device to actively pass air through the air sample container, and the passive method using PUF discs. The latter, also called the PUF-PAS method, has been widely used for organic pollutant measurements due to its advantages of low cost (no power supply required) and simple handling [61]. 

Three extraction techniques have been employed for the extraction of pesticides from this sampler (Figure 3): Soxhlet, SLE, and QuEChERS. In Soxhlet extractions, the filters and cartridges or PUF discs were washed with different organic solvents. The extraction process duration ranged from 4 h to 24 h. In the study carried out by Yu et al. [62], while PUFs were Soxhlet extracted, GFFs were cut and extracted by SLE with an *n*-hexane/acetone mixture in a microwave extractor. Similarly, in the studies of Ikenaka et al. [60] and Yera et al. [58], the filters were cut and sonicated with solvent mixtures of ethyl acetate/acetone (9:1, *v*/*v*) and ethyl acetate/acetonitrile (30:70, *v*/*v*), respectively. Lastly, Zhou et al. [59] developed a QuEChERS extraction to detect NEOs in PM_2.5_, using acetonitrile as the sorbent and applying a clean-up process with primary secondary amine (PSA). Subsequently, pesticides extracted from air samples were detected by LC–MS/MS and GC–MS. Specifically, Pirard et al. [63] split the final extract into two fractions for LC and GC analysis, in order to separate and detect volatile compounds more efficiently. In LC–MS/MS, the instruments operated in MRM mode with ESI^+^ (Pirard et al. [63] also worked in the negative ion mode). In GC–MS/MS analysis, the mass spectrometer operated with electron ionization (EI) or negative chemical ionization (NCI). Table 1 shows the analytical approaches and performance to detect pesticide levels in air (see also Appendix A, for more detailed information).

### 4.3. Analysis of Pesticides in Dust and Passive Samplers

Indoor dust, as well as road dust, can serve as a reservoir of semi-volatile organic compounds (SVOCs), including OCPs [43]. For this reason, pollution monitoring data of dust provide useful information on the behavior and fate of pesticides in the environment and can be utilized as another approach to study the exposure of these pollutants in urban environments. Our review includes three studies analyzing OCPs levels in dust. Figure 3 and Table 1 show the extraction techniques applied to dust samples and the main issues related to the analytical procedures. The research group of Yadav et al. [43,44] carried out two very similar studies in which they analyzed OCP (and PCB) residues in dust samples from different types of indoor environment, taken by vacuuming. For pesticide extraction, the freeze-dried and homogenized dust samples were extracted with dichloromethane using a Soxhlet extractor for 24 h, followed by silica–alumina column clean-up. In both studies, pesticide analyses were carried out by GC–triple quadrupole (QqQ)-MS, using the same capillary column and carrier gas. The mass spectrometer was operated operating using EI mode with selected ion monitoring (SIM). Anh et al. [34] screened 10 classes of micropollutants in road dust, including 10 OCPs. The samples were manually collected by sweeping the asphalt surface, and then homogenizing them into a representative pooled sample. The target analytes were extracted by SLE with acetone plus an acetone/hexane (1:1, *v*/*v*) solution, using an ultrasonic processor. Then, the extracts were purified by an activated silica gel column. Separation and detection of micropollutants were carried out by GC–MS equipped with an Automated Identification and Quantification System (AIQS-DB) system that facilitated the identification of the compounds.

To conclude the review on environmental monitoring, we include the study carried out by Arcury et al. [64], in which exposure to pesticides of children from rural and urban communities was evaluated under a different approach from those explained above. In practice, they gave the children silicone wristbands (with consent from parents or guardians) for passive exposure monitoring. The wristbands were cleaned after deployment with 18 MΩ·cm water and isopropanol to remove particulate matter, and then the analytes were extracted by SLE with ethyl acetate. Analytical interferences were removed by an SPE clean-up process using a silica column with acetonitrile. Recoveries between 14% and 142% were obtained from the QA/QC tests. Finally, the pesticide quantification was carried out by GC–ECD, using a DB-XLB column for confirmation. Children exposure was evaluated using information collected from an interviewer-administered questionnaire, comparing by statistical test the pesticide classes and levels detected with the participant personal and family characteristics. More information about this study is shown in Table 1.

### 4.4. Analysis of Pesticide Residues in Food

A total of 19 articles on the assessment of pesticide residues in food were selected. Foods analyzed are shown in Figure 4. A wide variety of vegetables were studied, including tomatoes, lettuce, kale, French beans, and water spinach. Specifically, in the study carried out by Yi et al. [65], 96 types of vegetables were analyzed including leafy vegetables, stem vegetables, roots, and tubers. Other foods analyzed in the articles reviewed were fruits (apples and peaches), cereals (wheat and maize, both straw and grain, as well as the maize corncob), different bee products, such as wax and honey, and even fish from estuaries, lagoons, or aquaculture. Lastly, Nougadère et al. [66] analyzed pesticide residues in other food products, including manufactured baby foods and common food such as cakes, pasta, or fried breaded fish. 

Three of these studies involved pesticide multiresidue analyses. Bommuraj et al. [67] screened hundreds of pesticide residues (over 600) of different classes in beeswax and honey, including NEOs, OPPs, OCPs, and pyrethroids (PYRs). Yi et al. [65] analyzed 283 different pesticide residues, including several insecticides, fungicides, herbicides, miticides, growth regulators, and one plant activator. In addition, Nougadère et al. [66] screened over 500 pesticides and metabolites in food composite samples, comprising pesticides of different chemical structures (mainly fungicides, insecticides, and herbicides). All the remaining studies evaluated specific groups of pesticides, such as the most used OCPs (e.g., DDT, heptachlor, endrin, and chlordane) or single compounds such as flumethrin, tebuconazole, tembotrione, or pymetrozine.
molecules-26-03688-t001_Table 1Table 1Selected analytical methods published between 2019 and 2021 for the analysis of pesticides in environmental matrices and food to analyze human exposure.SampleNº Pesticides or BiomarkersSample TreatmentSeparation and Detection TechniqueRecovery %LOD Ref.MatrixVolume/WeightMethodExtractionClean-upDrinking water (TPW and tap)50 mL7LLE30 mL DCMPurified passing sample through a chromatographic columnUPLC–QqQ-MS/MS: in MRMESI +73–94%30–70 ng/L[52]Drinking water (groundwater and tap)500 mL16SPEOasis HLB(6 cc/500 mg) eluted with 4 mL CAN + 4 mL methanol-UPLC–MS/MS: in MRMESI^+^74–123%0.01–0.2 ng/L *[53]Groundwater1000 mL56LLE20 mL DCM (×3): with three pH conditions (6.5–8.0, <2.0, and >10.0)-GC–MS: SIM and SCAN modeEI 70–133%2.5–247 (ng/L)[49]Tap water500 mL9LLE70 mL DCMSilica gel column with anhydrous Na_2_SO_4_ (CNWBOND 10 cc/10 g)GC–MS: in MRM mode EI 76–94%0.0011–0.43 ng/L[54]Surface water1000 mL65SPEC18 (6 cc/1000 mg)5 mL ethyl acetate + 5 × 2 mL DCMGC–MS/MS: in SRMEI --[50]Sediment and soil-PLEAcetone/DCM (1:1, *v*/*v*)Florisil ^®^ (6 cm^3^/1000 mg) eluted with 10 mL acetone/hexane (20/80 *v/v*)Surface water1000 mL8LLEEthyl acetate + methylene-GC–MS: in SIMEI80–94%1.05–2.60 ppb[55]Sediment5 gQuEChERS15 mL ACN 1% AA + 6 g MgSO_4_ + 1.5 g NaOAcd-SPE with 25 mg PSA + 25 mg C18 +7 mg GCB + 150 mg MgSO_4_Surface water10 mL224---DAI to LC–MS/MS: in MRMESI^+/−^34–135%1.0–106 ng/L[51]Sediment10 g119PLEDCM; purified in a Florisil^®^(6 cm^3^/1000 mg)Purification: 1º fraction: DCM + 50:50 DCM: ethyl acetate; 2º fraction: 20% DCM in hexane + 50% ethyl acetate in hexaneGC–MS/MS: in SIMEI 75–102%0.6–3.4 μg/kgSoil20 g21SoxhletAcetone/*n*-hexane (1:1, *v/v*)Florisil cartridge (6 cc/1000 mg) eluted with 5 mL *n*-hexane/acetone (95:5, *v/v*) GC–MS75.9–126.1%-[41]Soil5 g23QuEChERS10 mL ACN 1% AA + 1 g NaOAc + 4 g MgSO_4_dSPE with 50 mg PSA + 150 mg MgSO_4_
Polar compounds:LC–MS/MS:Ionization in + and −70–120%1–10 μg/kg[56]A-polar compounds:GC–MS/MSSoil1 g8SPE10 mL cartridge column packed with: Na_2_SO_4_ (0.5 g)Florisil (1 g, 60–100 mesh), acidic silica gel (1 g) + copper powder (0.5 g) eluted with 15 mL DCMGC–ECD: in SIMEI ionization80–96%0.001–0.025 ng/g[39]  Sediment 4 g8PLEAcetone/*n*-hexane (1:1, *v/v*); purified two times: (1) in an activated copper column (20% HCl); (2) ENVI-CARB/PSA cartridge (6 cc/500 mg)3 mL hexane(1st purification);6 mL hexane-ethyl-acetate (7/3, *v/v*) (2^nd^ purification)GC–MS: in SIMEI ionization89–118%-[40]Air particulates1344 m^3^46SoxhletHexane/acetone/MeOH (50:40:10 *v*/*v*/*v*)-UPLC–MS/MS: in MRMESI^+/−^72–128%0.04–0.1 ng/m^3^ *[63]GC–MS-QqQ: in MRMEI^−^Air particulates-7SLE10 mL ethyl acetate/acetone (9:1, *v/v*)-LC–MS/MS: in MRM modeESI^+^--[60]Air particulates30 m^3^10SoxhletAcetone Exchanged into hexane; purified in a silica gel columnGC–MS:EI ionization60–149%0.1–1 ng/m^3^[61]Air particulates432 m^3^26PUF: Soxhlet150 mL *n*-hexane/acetone (*v*/*v*, 1:1)Silica gel/alumina chromatographic column eluted with 70 mL DCMGC–MS: in SIM mode Negative chemical ionization (NCI)65–120%0.1–25.0 pg/m^3^[62]GFF: SLE25 mL *n*-hexane/acetone (*v*/*v*, 1:1)PM_2_._5_
158.4 m^3^4QuEChERS20 mL ACNdSPE with 0.4 g PSALC–MS/MS-QqQ: in MRMESI^+^78–97%0.0005–0.355 ng/m^3^[59]PM_2_._6_ and PM_10_
1627.2 m^3^34SLE500 µL ethyl acetate/ACN (30:70)-GC–MS: in SIMEI 90–144%0.14–0.44 ng/mL[58]Dust (indoor)10 g26SoxhletDCM; purified in a silica–alumina column-GC–MS-QqQ: in SIMEI 88–110%1.31–7.30 pg/g[43]Dust (indoor)20 g24Soxhlet300 mL DCM; purified in a silica–alumina column-GC–MS-QqQ: in SIMEI 88–110%1.31–7.30 pg/g[44]Dust (road)2 g10SLE10 mL acetone + 10 mL acetone/hexane (1:1, *v*/*v*); purified in a silica gel column DCM + hexane (purification)GC–MS60–120%0.0010–0.010 μg/g[34]Silicone wristband -75SLE50 mL ethyl acetate; purified in a C18 silica column (500 mg)9 mL ACN(purification)GC–ECD11–142% (median 55%)0.44–20.9pg/µL[64]Fish 3 g 8PLEAcetone/*n*-hexane (1:1, *v/v*)Twice:(1) with a copper column eluted with hexane(2) ENVI-CARB/PSA cartridge eluted with6 mL hexane/ethyl-acetate (7:3, *v/v*)GC–MS: in SIMEI 89–118%-[40] Fish3 g 18Soxhlet150 mL hexane/acetone (3:1, *v/v*)Glass column (30 cm × 1 cm) [1 g neutral alumina + 1 g neutral silica + 8 g acidified silica + 4 g Na_2_SO_4_] eluted with 50 mL DCM and hexane (1:1, *v/v*) GC–μECD61–136%0.0003–0.0054 ng/g[68]Cow´s milk2 g18LLE 3 mL *n*-hexane/DCM (1:1, *v/v*)Glass column (30 cm × 1 cm) [1 g 5% deactivated silica + 1 g 5% deactivated Florisil + 1 g Na_2_SO_4_] eluted with 15 mL *n*-hexane + 10 mL DCMGC–μECD70–109%0.003–0.63 ng/g[69]Wax20 g 1QuEChERS10 mL ACN; NaCl + MgSO_4_ + sodium citrate + sodium hydrogen citrate sesquihydratedSPE (150 mg MgSO_4_ + 25 mg C_18_ + 25 mg PSA)LC–MS/MS: in MRMESI^−^95%20 μg/kg *[70]Wax1 gMore than 600 QuEChERS10 mL water + 10 mL CAN + Supel™ QuE citrate/sodium bicarbonatedSPE using Supel™ QuE PSA/C18 clean-up tubeLC–MS/MS: in MRMESI^+/−^-0.0005–0.002 mg/kg[67] Honey2 g QuEChERS10 mL ACN + 4 g anhydrous MgSO_4_ + 1 g trisodium citrate dihydrate + 0.5 g disodium hydrogen citrate sesquihydrate + 1 g NaCldSPE clean-up with 900 mg anhydrous MgSO_4_ + 150 mg of PSAGC–MS/MS: in SRM modeEI ionization-0.002 mg/kgTomato10 g21QuEChERS10 mL ACN + 1 g NaCl + 1.5 g citratedSPE clean-up with 900 mg MgSO_4_ + 150 mg PSA 150 mg C_18_OPPs: GC–NPD72–116%0.5–10 μg/kg[71]Halogenated: GC–ECDLettuceMethyl-carbamates: HPLC–FLDImidacloprid and carbendazim: HPLC–DAD96 types of vegetables50 g283SLE100 mL ACN; 10 g NaClFor GC:Sep-Pak Florisil (6 cm^3^/1000 mg) eluted with 7 mL 20% acetone/hexaneFor LC:Sep-Pak NH_2_ (6 cm^3^/1000 mg) eluted with 5 mL 1% MeOH/DCMOPPs and nitrogen-containing compounds: GC–NPD82.5–103.1%0.0006–0.024 mg/kg[65]OCPs, dicarboximide and PYR: GC–μECDCarbamate pesticides: LC–FLDUV-detected compounds: LC–DAD: APCI^+^Tomato10 g 7QuEChERS

LC–MS/MS: ESI 76.84–96.32%0.10 μg/kg[72]French beans10 mL ACN + 150 mg MgSO_4_dSPE with 150 mg MgSO_4_ + 50 mg PSA + 50 mg GCBKale

Tomato10 g 2QuEChERS10 mL ethyl acetate + 4 g anhydrous MgSO_4_ + 1 g NaCldSPE 50 mg PSA + 150 mg anhydrous MgSO_4_GC–ECD83.1–102.2%0.01 mg/kg[73]Water spinach5 g2SLE10 mL ACN; 1.5 g NaCl- LC–MS/MS: ESI 91–101%0.02 mg/kg *[74]Chlorothalonil: GC–MS: in SRM modeEI 94–105%0.01 mg/kg *Kale10 g4SLE15 mL MeOHPurified with 50 mg C18LC–MS/MS-QqQ: in MRM modeESI^+/−^26.5–89.6%0.14–20.3 μg/kg[75]Apple10 g 3QuEChERS10 mL ACN + 1 g NaCl + 4 g MgSO_4_;dSPE with 250 mg MgSO_4_ + 100 mg PSA + 15 mg GCBLambda-cyhalothrin: GC/MS: in SIM modeEI 88–105%0.01 mg/kg *[76]Thiamethoxam and clothianidin: RRLC–MS/MS-QqQ: in MRM EI^+^Peaches10 g 2QuEChERS20 mL ACN + 3 g NaCldSPE with 100 mg C18, 100 mg PSA + 300 mg of MgSO_4_LC–MS/MS-QqQESI 83–119%0.01 mg/kg[77]Lettuce10 g18QuEChERS10 mL ACN/AA (99:1, *v*/*v*) + 6 g MgSO_4_ + 1.5 g NaOAc + 1.0 g sodium acetate trihydrate (CH_3_COONa·3H_2_O);d-SPE clean-up with 1.2 g MgSO_4_, 0.4 g C-18, 0.4 g PSA + 0.4 g Florisil GC × GC–TOF-MS-0.5–0.9 ng/g[78]Spinach74–106% Spring onions-Peanuts5 g73–101% Lettuce2 g8SLE10 mL ACN + 4 g Na_2_SO_4_ +1 g NaCl;Two SPE purification: (1) d-SPE clean-up with 75 mg of C18, 75 mg of PSA + 1350 mg of Na_2_SO_4_; (2) SPE cartridges of 6 cc/100 mg eluted with 4 mL ethyl acetate GC–MS/MS-QqQ: in SRM mode EI -0.013–4.45 µg/kg [79]Tomatoes69–96%Cauliflower47–87%Broad beans41–98%Wheat grain5 g2 QuEChERS10 mL ACN + 1 g NaCl + 4 g MgSO_4_; d-SPE clean-up with 150 mg MgSO_4_ + 50 mg C18+ 10 mg GCBHPLC–MS/MS: in MRM mode 87–112% (epoxiconazole) and 85–102% (pyraclostrobin)0.01 mg/kg *[80]Wheat straw1 g Maize grain5 g2 QuEChERS 10 mL 5% AA/ACN + 1 g NaCl + 4 g MgSO_4_;Two types of dSPE: Maize grain and straw: 50 mg PSA + 5 mg MWCNTs + 150 mg MgSO_4_; Corncob extract: 50 mg PSA + 150 mg MgSO_4_HPLC–MS/MS: in MRM modeESI^−^98–107% (tembotrione) and 90–108% (M5)0.43–1.5 μg/L[81]Maize corncob2 gMaize straw 1 g Soybean 5 g5QuEChERS10 mL 1% AA/ACN + 1 g NaCl + 3 g MgSO_4_;Two types of dSPE: Soybean: 50 mg C_18_ + 150 mg MgSO_4_;Green soybean and straw: same + 5 mg MWCNTsUPLC–QqQ-MS/MS:ESI 71–116%0.018–0.125 μg/kg [82]Green soybean5 g Soybean straw2.5 gCommon food (vegetables, fruit, cakes)10 g (non-cereal-based)5 g (cereal-based)516Non-cereal-based: QuEChERS (vers. 1);Cereal-based: QuEChERS (vers. 2) Vers. 1: 10 mL ACN + 1 g NaCl + 4 g MgSO_4_ + 0.5 g disodium hydrogen citrate sesquihydrate + 1 g trisodium citrate dihydrateVers. 2: 20 mL ACN + 1 g NaCl + 4 g MgSO_4_ + 0.5 g disodium hydrogen citrate sesquihydrate + 1 g trisodium citrate dihydrate; dSPE: 150 mg MgSO_4_ + 25 mg PSA221 analytes: LC–MS/MS-QqQ: ESI^+/−^70–120%0.1–10 µg/kg[66]Baby food (prepared)Non-cereal-based:QuEChERS (vers. 3);Cereal-based: QuEChERS (vers. 4)Vers. 3: 10 mL ethyl acetate + 1 g NaCl + 4 g MgSO_4_ + 0.5 g disodium hydrogen citrate sesquihydrate + 1 g trisodium citrate dihydrate;; Vers 4: 20 mL ethyl acetate/cyclohexane 81:1 + 1 g NaCl + 4 g MgSO_4_ + 0.5 g disodium hydrogen citrate sesquihydrate + 1 g trisodium citrate dihydratedVers.3: Purified in HPGPC columnVers 4: dSPE with 25 mg PSA + 25 mg C18 + 5 mg carbon135 analytes: GC–MS/MS: in MRM mode EI* LOQ value was reported, when LOD was not available. µECD: microelectron capture detector; AA: acetic acid; ACN: acetonitrile; APCI: atmospheric pressure chemical ionization; DAD: diode array detection; DAI: direct aqueous injection; DCM: dichloromethane; ECD: electron capture detection; EI: electron ionization; ESI: electrospray ionization; FA: formic acid; FLD: fluorescence detector; GCB: graphitized carbon black; GFF: glass fiber filter; LLE: liquid–liquid extraction; MRM: multiple single-reaction monitoring; MWCNTs: multiwalled carbon nanotubes; NaOAc: sodium acetate; NCI: negative chemical ionization; NPD: nitrogen–phosphorus detector; OCPs: organochlorine pesticides; OPPs: organophosphorus pesticides; PM: particulate matter; PSA: primary secondary amines; PLE: pressurized liquid extraction; PUF: polyurethane foam; PYR: pyrethroid; SIM: selected ion monitoring; SLE: solid–liquid extraction SPE: solid-phase extraction; SRM: selected reaction monitoring; TRV: toxicological reference value; Water: surface water.

Pesticide residues in food were extracted by QuEChERS, SLE, LLE, PLE, and Soxhlet. QuEChERS procedures were quite similar each other and required a sample amount between 1 and 20 g. Acetonitrile was by far the most used solvent, followed by ethyl acetate. Extractions were followed by clean-up process, mainly carried out by d-SPE with magnesium sulfate, C-18, and PSA (primary secondary amine). A distinction between cereal- and non-cereal-based food was carried out by Nougadère et al. [66], with minor variations in the extraction protocol (eluent volume, subsequent cleaning process, etc.). In SLE, the sorbents used were acetonitrile and methanol. Sodium chloride was added to the sample in acetonitrile separations to get the salting out effect. Subsequently, the extractions were accelerated using a high-speed homogenizer, by sonication, or by vortex agitation. The extracts were mainly cleaned up using C18, PSA, or sodium sulfate, or using commercial SPE cartridges (of 6 mL and 100–1000 mg), with different eluents such as ethyl acetate. Pesticide residues in milk were extracted after acidification with formic acid by a LLE with *n*-hexane/dichloromethane aided by a vortex [69]. The extracts were cleaned by passing through a silica column (with Florisil and anhydrous sodium sulfate) and eluting the pesticides with *n*-hexane followed by dichloromethane. PLE was performed on fish samples with a mixture of acetone/*n*-hexane (conditions: 125 °C and 1500 psi) [40]. Then, two purifications were performed. First, the fish extract was treated with sulfuric acid and passed through a prerinsed glass tube with acid/silica, eluting with hexane. Subsequently, the eluate was cleaned in an ENVI-CARB/PSA cartridge and eluted with hexane/ethyl acetate. Lastly, Olisah et al. [68] followed a different methodology for the pesticide extraction from fresh fish samples, using a hexane/acetone mixture in a Soxhlet extractor for 24 h. After the extract was reduced, pesticide residues were eluted with dichloromethane and hexane in a glass column previously prepared with neutral alumina, neutral and acidified silica, and sodium sulfate. 

Pesticide detection and quantification were performed using different apparatus and techniques. LC–MS/MS with a triple quadrupole (QqQ) was used to analyze wax samples and some vegetable and fruit samples, operating in MRM mode. ESI ionization differed between positive and negative depending on the target analytes. Fan et al. [76] also applied LC–MS/MS-QqQ with positive electron ionization to specifically analyze thiamethoxam and clothianidin compounds. LC–FLD (fluorescence detector) was used to analyze residues of carbamate pesticides, and LC–DAD (diode array detection) was used to analyze imidacloprid and carbendazim from vegetables. GC–MS/MS was widely used in multi-analysis, working in SRM, MRM, and SIM modes. Buah-kwofie et al. [78] analyzed OCP residues from vegetable samples by two-dimensional gas chromatography/time-of-flight-mass spectrometry (GC × GC–TOF-MS). In addition, GC–microeletron captutre detection (µECD) was used in several studies to analyze OCPs, OPPs, and PYRs in food samples of different classes (milk, fish, and vegetables). Lastly, GC–nitrogen phosphorus detection (NPD) was used to analyze some OPP analytes and other nitrogen-containing compounds from vegetable samples. All information regarding the analytical procedures, equipment, recoveries, and detection limits determined are summarized in Table 1. The stationary and mobile phases can be found in Appendix A).

## 5. Direct Approaches

### 5.1. Analysis of Pesticides and Their Metabolites in Human Biomonitoring 

This review includes 30 studies on human biomonitoring plus two studies in which urine and atmospheric particulate matter were analyzed in parallel. The biological matrices studied are shown in Figure 5. Fifty-seven percent of the articles analyzed the levels of pesticide excreted in urine. Other biological fluids analyzed included serum, blood, and plasma. In addition, some studies were focused on assessing pesticide exposure to pregnant and lactating women and its correlation with possible prenatal exposure to the fetus or to the newborn through breastfeeding [83,84,85,86]. For this purpose, breast milk, cord blood, and placenta samples were analyzed. Only one study measured contaminant levels in hair in addition to urine samples [87]. Hair showed advantages over urine as it was easier to collect, handle, and store, and it allowed the assessment of the cumulative exposure to pesticides.

The target analytes included a wide spectrum of pesticide classes such as OPPs, OCPs, PYR, NEOs, and carboxamides, as well as some of their specific and nonspecific biomarkers (e.g., dialkyl phosphate (DAP) metabolites, *m*-PYR, and *m*-NEOS, among others). About 78% of the biomonitoring studies analyzed fewer than 20 pesticides or metabolites. Many of these involved the exposure assessment of a defined population to a particular compound by detecting the compound in question or its most common metabolites. For example, Stajnko et al. [88] estimated the exposure to glyphosate (GLY) and its major metabolite aminomethylphosphonic acid (AMPA) in a young population of an agricultural community. The remaining studies evaluated between 26 and 43 substances except the study carried out by Smadi et al. [85], in which 161 pesticides (and three antibiotic families) were tested in breast milk from lactating women living in refugee camps.

In relation to the extraction techniques, most of the analyses showed a similar scheme, as summarized in Figure 5. In 57% of the studies, prior to extraction, enzymatic or chemical digestion was carried out to hydrolyze the phase II conjugates (e.g., glucuronides, sulfates). Enzymatic digestion was mainly performed in urine or serum by addition of a β-glucuronidase-buffered solution [89]. In other studies, deconjugation of metabolites in urine samples was carried out using formic or hydrochloric acids. These acid solutions were also applied in other analyses to denature plasma proteins in blood matrices [90]. Biomarkers were subsequently extracted from the biological samples by LLE, SPE, QuEChERS, SLE, and PLE. In LLE, a variety of sorbents, such as acetonitrile, ethyl acetate, and methyl *tert*-butyl ether (MTBE), among others, were reported. In some cases, two or three consecutive extractions were performed, followed by a clean-up process by d-SPE. In SPE methods, the analytes were extracted mainly by Oasis^®^ HLB cartridges with different bed weights and column capacities. Oasis WAX or Strata-X-AW cartridges were also reported. The QuEChERS extraction followed the characteristic scheme including a clean-up process through d-SPE. For hair analysis, Hernández et al. [87] extracted dialkyl phosphate (DAPs) metabolites by SLE, using methanol and purifying the extract in a test tube containing potassium carbonate and sodium disulfate. Prior to the last evaporation, in which the residue was reconstituted in toluene, the extract was incubated at 80 °C with potassium carbonate, acetonitrile, and pentafluorobenzylbromide (PFBBr) as a derivatization agent. Lastly, Bassig et al. [91] analyzed serum concentrations of 11 OCPs and their metabolites using PLE. In brief, previously serum samples were freeze-dried into 22 mL extraction cells and extracted at 100 °C and 1500 psi using 20% dichloromethane in hexane in the static mode. 

The target compounds extracted from biological matrices were detected and quantified by five different techniques. Analytes excreted in urine were analyzed mainly by LC–QqQ-MS/MS and by GC–QqQ-MS/MS, both in MRM mode. ESI for LC and EI for GC were reported. As an exception, Papadopoulou et al. [92] detected urine metabolites using a UPLC system coupled with a QTOF. GC–MS was also applied to detect pesticides and metabolites using EI or ECNI in blood, plasma, breast milk, hair samples, and cord blood and GC–ECD to detect pesticides in breast milk. HRGC–HRMS was used to detect analytes from serum matrix. The main characteristics of the analytical procedures for the biological matrices, including results of QA/QC tests, are detailed in Table 2. Additional information is shown in Appendix A).

### 5.2. Analysis of Human Biomarkers through Wastewater-Based Epidemiology

Wastewater-based epidemiology (WBE) is a recent approach to human biomonitoring based on the measurement of human biomarkers in urban wastewater and the subsequent calculation using the concentrations of biomarkers detected to determine population exposure or consumption [93,94]. This method assesses spatial and temporal trends and responses to specific events within a study catchment [95]. It is a noninvasive technique for the population and only requires a few 24 h composite samples of the influent to be collected and analyzed [93]. Today, WBE is an essential tool to estimate consumption of illicit and licit drugs in the population, as well as to monitor human exposure to some common contaminants, such as pharmaceuticals, plasticizers, flame retardants, or pesticides [95,96]. In relation to pesticide exposure, these studies are based on the selection and measurement of suitable urinary metabolites that end up in the sewer system. Human metabolites of pesticides, other than the starting product, which were stable in sewage system waters over a long period were specifically selected. Some metabolites were not specific to a particular pesticide but to a group and, as such, could only be used to assess exposure to this group (e.g., pyrethroids, and organophosphates) [95]. As this is a relatively young and growing field, only four studies that applied WBE to assess the pesticide exposure of a population were found in our search. Table 2 shows the main characteristics of the analytical techniques applied. 

Kasprzyk-Hordern et al. [45] carried out a complex study investigating biomarkers of several groups of pollutants in untreated wastewater from five cities in southwest England on a large scale. Nine fungicides and herbicides were studied along with other compounds such as pharmaceuticals, illicit drugs, lifestyle chemicals, and personal care products. Rousis et al. [46] assessed the levels of 14 biomarkers including triazine, pyrethroid, and organophosphate (OPP) metabolites in untreated wastewater from four Norwegian cities. The study carried out by Devault et al. [47] included the analysis of 18 biomarkers of pesticides present in untreated wastewaters from Martinique (French West Indies), belonging to the groups of triazines, pyrethroids, and OPPs. Lastly, Devault et al. [48] also evaluated the presence of chlordecone, an organochlorine insecticide, as well as its subproducts chlordecol and 5b-hydrochlordecone (CLD5BH) in untreated wastewaters. SPE was the extraction technique used by the three studies. Rousis et al. [46] also analyzed alkyl phosphate compounds by direct injection. The stationary phase and the eluent solution used in all cases were Oasis ^®^ HLB and methanol, respectively. The analytes extracted by SPE were analyzed by LC and UHPLC–QqQ-MS/MS systems mostly in positive ionization mode. Kasprzyk-Hordern et al. [45], in addition to ESI^+^, tested ESI^−^ using a mobile phase consisting of water/methanol; however, in this case, NH_4_F was used instead of acetic acid to favor the ionization of negatively charged molecules. Furthermore, Devault et al. [47] determine metabolites of chlordecone by LLE extraction, adding NaCl to the sample and using a mixture of hexane/acetone (85/15, *v/v*). Extracts were analyzed by GC–QqQ-MS/MS since these compounds are nonpolar and volatile. All the methods described were previously validated in terms of scope, specificity, accuracy, sensitivity, and repeatability, with recoveries around 80–120% and a sensitivity of ng/L, with the latter being higher for LC analysis.
molecules-26-03688-t002_Table 2Table 2Analytical methods published between 2019 and 2021 for the analysis of pesticides in biological matrices and between 2018 and 2021 for the wastewater-based epidemiology (WBE) studies to analyze human exposure to pesticide residues.SampleNº Pesticides or BiomarkersExtractionSeparation and Detection TechniqueRecovery %LOD Ref.MatrixVolume/WeightMethodPretreatment/Other FeaturesExtraction/Clean-UpUrine3 mL5LLEEnzymatic digestion prior to extraction (175 μL β-glucuronidase enzyme + 125 μL 0.1 mol/L HAC–NaAC buffer)2 mL ethyl acetateHPLC–MS/MS-QqQ: in MRMESI^+^71–107%0.005–0.02 ng/mL *[97]Urine4 mL26LLE:dialkyl phosphates (DAPs)Digestion prior to extraction(800 µL HCl 6 M)4 mL ethyl acetate + 4 mL diethyl etherLC–MS/MS: in SRM ESI^−^ and APCI^−^70–120%0.25–0.50 ng/mL *[98]QuEChERS:specific metabolitesEnzymatic digestion prior to extraction (10 μL of β-glucuronidase aryl sulfatase enzyme)10 mL ACNUrine2 mL6LLE-1º LLE: 2 mL ACN + 2 mL diethyl ether2º LLE extraction: 5 mL water + 5 mL n-hexaneGC–MS/MSEI^+^92–118%0.01–0.1 ng/mL[99]Urine2 mL11LLE-2 mL ACN + 2 mL diethyl etherGC–MS: in MRMEI^+^75–100%0.1–0.5 ng/mL[100]Urine1 mL9SPEEnzymatic digestion prior to extraction (250 μL of β-glucuronidase enzyme in ammonium acetate buffer (10 mM, pH 6.7))OASIS HLB(6 cc/150 mg) eluted with 6 mL MeOHLC–MS/MS: in MRMESI^−^43–100%0.001–0.3 ng/mL[101]Urine1 mL3SPEEnzymatic digestion prior to extraction (750 μL of β-glucuronidase buffer solution)OASIS HLB(6 cc/150 mg) eluted with 750 μL acetoneHPLC–MS/MS-QqQ: ESI^+/−^80–120%0.003–0.4 ng/mL[89]Urine4 mL26LLE:dialkyl phosphates (DAPs)Digestion prior to extraction(800 µL HCl 6 M)4 mL ethyl acetate + 4 mL diethyl etherLC–MS/MS: in SRMESI^−^82–117%0.125–5.0 ng/mL *[102]5 mLQuEChERS:specific metabolitesEnzymatic digestion prior to extraction (10 μL of β-glucuronidase aryl sulfatase enzyme)10 mL ACN60–120%Urine0.1 mL2LLE-1 mL ACNGC–MS/MS:ESI^−^90–110%0.1 ng/mL *[88]Urine1 mL3LLE-1 mL ethyl acetateUPLC–MS/MS: in MRMESI^−^86–108%0.02–0.09 ng/mL[103]Urine4 mL43SPE:parent pesticides and desethylterbuthylazineEnzymatic digestion prior to extraction (20 μL of β-glucuronidase enzyme + 20 μL sulfatase + 2 mL 0.2 M sodium acetate buffer)Oasis HLB(6 cc/200 mg) + Chromafix dry sodium sulfate cartridge eluted with 4 mL hexane:DCN 95:5 (v/v)GC–MS/MS-QqQ: in MRMPCI^−^-0.05–0.92 ng/mL *[63]Enzymatic digestion prior to extraction (20 μL of β-glucuronidase enzyme + 20 μL sulfatase + 2 mL 0.2 M sodium acetate buffer)Oasis HLB(6 cc/200 mg) eluted with 3 mL DCMLC–MS/MS: in MRMESI^+^-2 mLSPE: DAPsDigestion prior to extraction(300 μL HCA 3 M);Oasis WAX(3 cc/60 mg) eluted with 5% ammoniac in MeOHGC–MS-QqQ: in MRMPCI^−^-3 mLLLE:pyrethroid metabolites and other biomarkersEnzymatic digestion prior to extraction (20 μL of β-glucuronidase enzyme + 20 μL sulfatase + 2 mL 0.2 M sodium acetate buffer)4 mL diethyl ether + 2 mL sodium dihydrogen phosphate (0.2 M)GC–MS-QqQ: in MRMEI^+^-Urine3 mL6LLEEnzymatic digestion prior to extraction (0.3 mL 1.0 M ammonium acetate with 66 units of β-glucuronidase enzyme)4 mL ethyl acetateHPLC–MS/MS: in MRM modeESI^+^76–107%0.0002–0.006 ng/mL[104]Urine1 mL2SPEEnzymatic digestion prior to extraction (10 μL IS solution (1 mg/L) and 100 μL β-glucuronidase enzyme (124 units/mL) solution with ammonium acetate)PEP cartridges eluted with 1 mL ACNLC–MS/MS: in MRM modeESI^+^78–111%0.029–0.038 ng/mL *[105]Urine0.5 mL11SPEEnzymatic digestion prior to extraction (400 μL 0.2 M sodium acetate with 745 units/mL of β-glucuronidase enzyme and 56 units/mL of sulfatase)Oasis^®^ HLB(3 cc/60 mg) eluted with 3 mL acetone + 3 mL hexaneHPLC–MS/MS-QqQ: ESI^+/−^84–115%0.025–0.05 ng/mL[106]Urine1 mL6SPEEnzymatic digestion prior to extraction (750 μL β-glucuronidase enzyme buffer solution)Quadra 3 Liquid Handling Station and OASIS HLB 96-well (automated SPE) eluted with 750 μL acetone (in two 325 μL aliquots)HPLC–MS-QqQ: in SRM modeESI^+/−^90–110%0.1–0.5 µg/L[107]Urine0.6 mL12-Digestion prior to extraction (25 μL of 1.3% FA in water)-LC–MS/MS-QqQ: ESI-0.1–0.5 ng/mL[108]Urine2 mL6LLE-2 mL ACN + 2 mL diethyl etherGC–MS/MS-QqQ: EI76–110%0.0032–0.31 ng/mL[109]Urine2 mL2LLEDigestion prior to extraction (0.5 mL concentrated HCl)2 mL MTBEGC–MS/MS-QqQ: in MRM modeEI91–109%0.049–0.075 µg/L[110]Urine1 mL7SPE-Presep RPP cartridges (60 mg) + ENVIcarb/PSA (500 mg/300 mg) eluted with 8 mL DCM:ACN (2:8; *v/v*)LC–MS/MS: in MRM modeESI^+^96–102%0.05–0.2 ng/mL *[60]Urine0.3 mL3SPEDigestion prior to extraction (40 μL FA)Strata-X-AW eluted with 0.5 mL acetone (5% TEA)UHPLC–TOFMS: ESI^−^42–108% (100 ng/mL)0.05–3.03 µg/L[92]Blood2–5 g6SPEDigestion prior to extraction (5 mL FA:2-propanol (4:1, *v*/*v*) and diluted with 5 mL 10% 2-propanol in water)ASPEC XL4 + Oasis © HLB (3 cc/400 mg) (automated SPE)GC–MS/MS-QqQ-0.3–1.52 pg/gUrine5 mL4SPE-C-18 Sep-Pak cartridges (500 m eluted with 5 mL DCMUHPLC–MS/MS-QqQ: in MRM mode ESI^+^60–120%0.5–1.0 µg/L[87]Hair (dried)0.05 gSLEIncubated in ultrasonic bath (4 h) with 2 mL MeOH prior to extraction2 mL MeOH and cleaned up in an econofilter with 15 mg of K_2_CO_3_ + 50 mg Na_2_S_2_O_5_ eluted with 1 mL ACN + 15 mg K_2_CO_3_ + 0.1 mL PFBBRr in ACN (1/3, *v/v*))GC–MS: in SIM mode75–107%3–6 pg/gUrine1 mL11SPE:3-phenoxybenzoic acid (nonspecific metabolite of PYR)Enzymatic digestion prior to extraction (750 μL of β-glucuronidase buffer solution)OASIS HLB(6 cc/150 mg) eluted with 750 µL acetoneHPLC–MS/MS-QqQ: ESI^+/−^80–120%0.015–4.0 ng/mL[111]2 mLLLE: dialkyl phosphates (metabolites of OPPs)-2 mL ACN + 2 mL ethyl etherGC–MS/MS: in MRMESI^+^75–100%Serum4 gSPE: OCPs-Oasis HLB (540 mg) (automated SPE workstation); Clean-up with two-layered SPE cartridge eluted with 12 mL DCM; 8 mL hexaneHRGC–IDHRMS: ESI^−^69–98%Serum0.5 mL26SPE-Oasis HLB (6 cc/500 mg);Clean-up with a small multilayer silica gel column (2 mL, 1.5 g) 10 mL n-hexane; 7.5 mL hexaneHRGC–HRMS:EI^+^30–124%0.07–13.44 pg/mL[112]Serum0.250 mL31SPEEnzymatic digestion prior to extraction (400 μL 0.2 M sodium acetate buffer with 745 units/mL of β-glucuronidase enzyme + 56 units/mL of sulfatase)Oasis HLB (3 cc/60 mg) eluted with 3 mL acetone + 3 mL hexaneHPLC–MS/MS: ESI^+^80–119%0.001–1.46 ng/mL[113]Serum5–10 mL (collected)8SPE-Sepra C18-E with Silica gel/Sulfuric Acid (2:1 *w/w*)HRGC–HRMS: in SIM mode-0.001–0.005 ng/mL[114]Serum1 g11PLEDried serum in extraction cells with 3 g hydromatrix20% DCM in hexaneGC–HRMS: EI-5 pg/g[91]Serum2 g9LLEDigestion prior to extraction (0.5 mL 6 M HCl)2.5 mL isopropanol + 6 mL of 50% MTBE in hexaneTwo clean-up cartridges: top: 0.2 g silica gel; lower: 1.0 g 33% sulfuric acid in silica gel (v/v) eluted with 10 mL 5% DCM in hexaneGC–IDHRMS64–74%11.5 ng/g[115]Plasma0.5 mL1LLEDigestion prior extraction (2 mL FA (50% *v/v*))5 mL hexane; Purified with ISOLUTE Florisil cartridges: 0.5 g anhydrous sodium sulfate + 1.8 g acidified silica eluted with 20 mL hexane/DCM (19:1 *v/v*)GC–MS/MS: in MRM modeEI-0.01 ng/mL[90]Breast milk2 mL18LLE-First LLE: 15 mL hexane/acetone (1:1);Second LLE: 10 mL hexane/acetone;Third LLE: 15 mL sodium sulfate 2% + 10 mL hexane + 5 mL acetoneGC–MS60–120%1.7–4.3 ng/g[83]Breast milk10 mL3QuEChERS-20 mL ACN + 4 g anhydrous MgSO_4_ + 1.5 g anhydrous NaCl; d-SPE clean-up with 50 mg PSA + 50 mg C18 + 750 mg MgSO_4_Analyzed with GC–ECD; confirmed with GC–MS85.8–120%0.005–0.05 mg/kg[84]Breast milk10 mL161QuEChERS-10 mL ACN + 4 g MgSO_4_ + 1 g NaCl + 1 g sodium dibasic citrate + 0.5 g sodium tribasic citrate; Clean-up in Polish EMR lipid tube (MgSO_4_ + NaCl)LC–MS/MS: in MRM modeESI80–120%5 µg/kg *[85]Breast milk1–5 g28LLE-5 mL cyclohexane/acetone (3:2); One aliquot + 97.5% H_2_SO_4_One aliquot + GPCHRGC–ECD77–159%0.002–0.041 ng/g[86]Maternal bloodPlacentaHRGC–LRMS: in SIM modeECNI104–161%0.014–0.626 ng/gCord bloodUntreated wastewater50 mL14Alkyl phosphates: direct injection; Others: SPEFinal extract reconstituted in 100 μL MilliQ-waterOASIS^®^ HLB (3 cc/60 mg) eluted with 3 mL MeOHLC–QqQ-MS/MS: in SRMIonization (+ and −)80–120%0.30–474 ng/L[46]Untreated wastewater50 mL9SPEFinal extract reconstituted in 500 µL 80:20 H_2_O:MeOHOASIS^®^ HLB (3 cc/60 mg) eluted with 4 mL MeOHUHPLC–QqQ-MS/MS: in MRMIonization (+ and −)80–120%0.02–0.95 ng/L[45]Untreated wastewater50 mL18SPEFinal extract reconstituted in 100 μL of MilliQ-waterOASIS ^®^ HLB (3 cc/60 mg) eluted with 3 mL MeOHLC–MS/MS-QqQ: in SRM modeIonization (+ and −)75–115%1.0–790 ng/L[47]Untreated wastewater100 mL3LLEFinal extract 1 mL hexane/acetane concentrated to 0.5 mL30 mL hexane/acetane (85:15, *v/v*)GC–QqQ-MS/MS: Ionization (+)80–120%100 ng/L[48]* LOQ value was reported, when LOD was not available. AA: acetic acid; ACN: acetonitrile; ASE: accelerated solvent extraction; BSTFA: *N*,*O*-bis(trimethylsilyl)trifluoroacetamide; CIP: chloroiodopropane; DAP: dialkyl phosphates; DCM: dichloromethane; ECNI: electron capture negative ionization; EMR: enhanced matrix removal; ESI: electrospray ionization; FA: formic acid; GPC: gel permeation column; HAC/NaAc: acetic acid/sodium acetate; HCA: hydrochloric acid; ID: internal diameter; LLE: liquid–liquid extraction; MRM: multiple reaction monitoring; MTBE: methyl *tert*-butyl ether; MTBSTFA: *N*-*tert*-butyldimethylsilyl-*N*-methyltrifluoroacetamide; NH_4_OAc: ammonium acetate; OCPs: organochloride pesticides; OPPs: organophosphate pesticides; PFBBr: pentafluorobenzyl bromide; PLE: pressurized liquid extraction; PYR: pyrethroids; SIM: selected ion monitoring; SLE: solid–liquid extraction; SPE: solid-phase extraction; SRM: selected reaction monitoring; TEA: triethylamine; TPAF: tripropylammonium formate.

## 6. Conclusions

Extraction techniques reported in environmental matrices analyses were LLE, SPE, PLE, QuEChERS, and Soxhlet. QuEChERS was by far the most widely applied extraction methodology in the food samples. In biological matrix analyses, extraction techniques were dominated by LLE and SPE. Although the procedures for sample preparation and preconcentration of analytes mostly followed a common scheme, a wide variety of techniques and modifications of these were observed, especially in water, soil, and sediment analyses. In relation to quantification techniques, the potential of LC–MS/MS and GC–MS/MS is unquestionable. In addition, other techniques were reported such as LC–FLD in food or GC–µECD in biological sample analyses. Exposure and risk assessment most widely involved the determination of EDI and the hazard quotient (HQ), although several alternative consumption, exposure, and risk indices were reported. Therefore, this review included the most recommended techniques for each type of study to be carried out: monitoring of environmental impact due to pesticide pollution in a study area and relationship with human health, assessment of agricultural practices and human exposure through food consumption, and individual and direct exposure analysis. 

Alternative environmental samples such as dust accumulated indoors or in open spaces were reported, allowing another interesting pathway to assess exposure. In addition, the use of passive samplers on study individuals may be interesting as a more accurate technique to measure the actual exposure of the population, although optimization of the technique is required. Vegetables and fruits were the most widely analyzed food matrices and were demonstrated to be effective in the analysis of dietary exposure. The requirements for sample handling, storage, and preparation, as well as the limitations related to the time period in which exposure is analyzed (usually last 24 h), must be taken into account in biomonitoring studies. For this reason, it is interesting to consider other alternatives such as hair analysis, which allows the assessment of cumulative exposure, or less invasive and sensitive techniques such as WBE, which also allows the assessment of a larger population size to carry out studies on a larger scale.

In conclusion, all these studies have contributed to an important advance in the knowledge of analytical techniques for the detection of pesticide levels and the subsequent assessment of nonoccupational human exposure. A gap identified in this review is the large number of methodologies reported, which require consensus to optimize protocols and to allow larger-scale, multi-analysis studies at both regional and national level. It is expected that, soon, knowledge in these fields will increase.

## Figures and Tables

**Figure 1 molecules-26-03688-f001:**
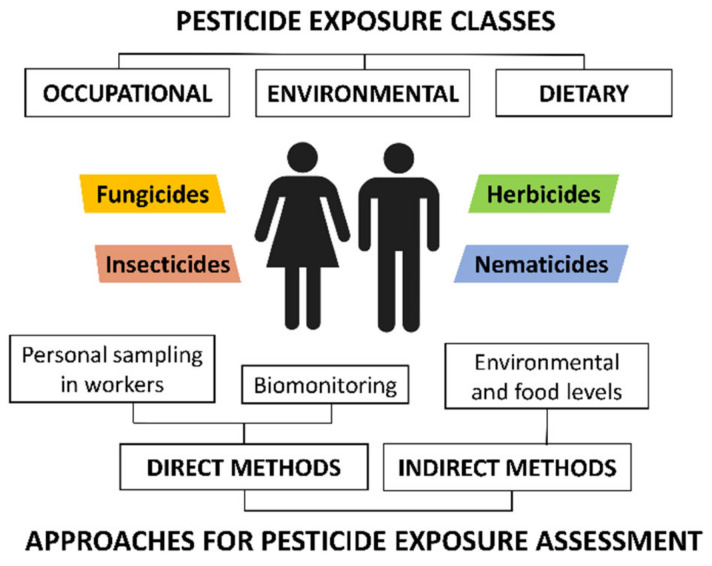
Classification of pesticide exposure routes and methodologies for pesticide exposure assessment.

**Figure 2 molecules-26-03688-f002:**
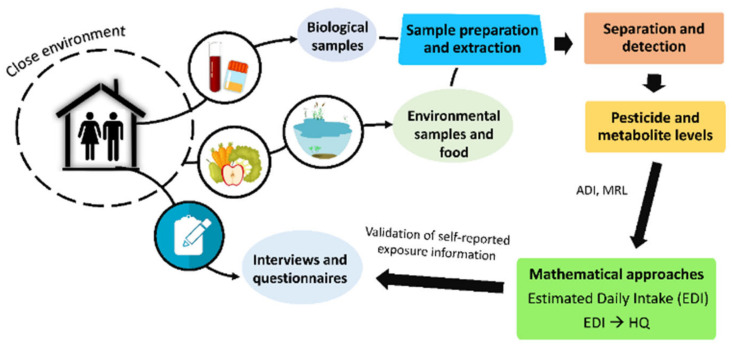
Main steps involved in a comprehensive assessment of human exposure to pesticides.

**Figure 3 molecules-26-03688-f003:**
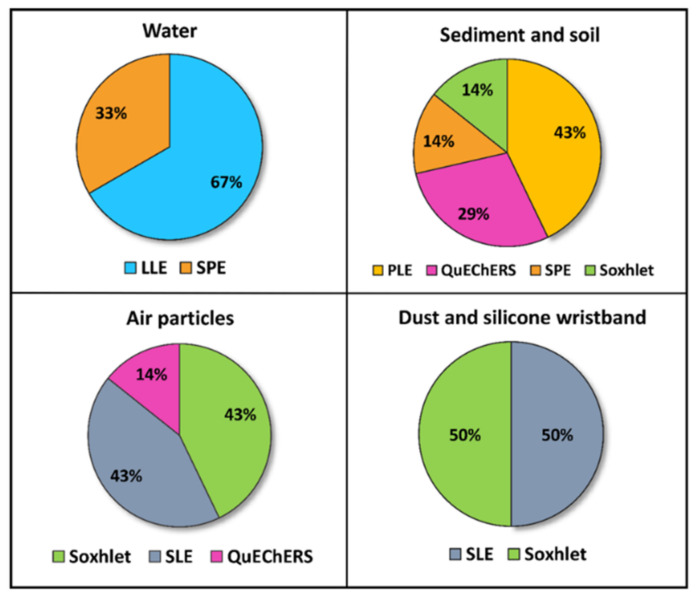
Percentage of articles (2019–2021) according to the extraction procedures applied in every environmental matrix studied (Source: Table 1). LLE: liquid–liquid extraction; SPE: solid-phase extraction; PLE: pressurized liquid extraction; QuEChERS: Quick, Easy, Cheap, Effective, Rugged, and Safe extraction method.

**Figure 4 molecules-26-03688-f004:**
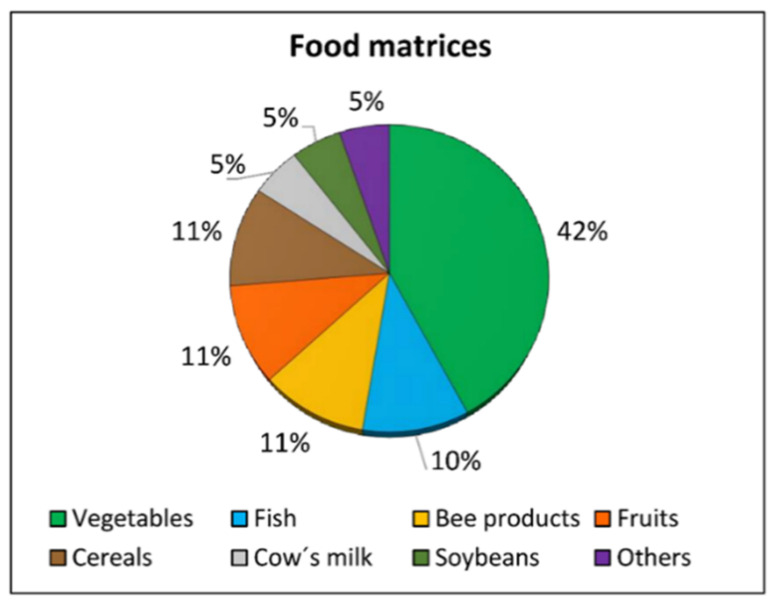
Food matrices according to the percentage of studies (2019–2021) that evaluated the levels of pesticide residues in them and the associated risk of exposure of the population (Source: Table 1).

**Figure 5 molecules-26-03688-f005:**
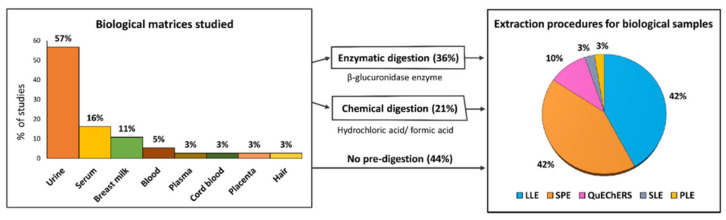
Percentage of studies (2019–2021) according to the biological matrices analyzed and the methodologies followed for sample preparation and subsequent extraction (Source: Table 2). LLE: liquid–liquid extraction; SPE: solid-phase extraction; QuEChERS: Quick, Easy, Cheap, Effective, Rugged, and Safe extraction method; SLE: solid–liquid extraction; PLE: pressurized liquid extraction.

## Data Availability

Data sharing not applicable.

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
