# Peer review of "Non-Occupational Exposure to Pesticides: Experimental Approaches and Analytical Techniques (from 2019)"

_molecules, 2021, doi:10.3390/molecules26123688_

Round 1

Reviewer 1 Report

L8 – delete „.“

Figure 1 is strange and should be adapted/simplified/splitted for the readers

Figure 3, Fig 4, Fig 5 – no references are included

Table 1 should be simplified, the clan-up part can be deleted

In Table2 samples can be divided under one name, not to repeat itself constantly. Volumes can be deleted.

The manuscript sounded interesting at first, but after closer examination, I found it difficult to follow. The reference were not mentioned in figures. Tables are extensive and should be simplified. There is no need to repeat information in the text and in tables.

I would expect more systematical approach to the topic, without repeating. The time frame of the review should be mentioned in the title (from 2018). In general, the title should be more focused on what the manuscript is about.

Supplementary material is hard to follow, especially tables S1 and S2 with all this formulas. Could this be simplified or removed?

In general, the manuscript is not bad, but it is confusing with potentialy overloaded infos.

Author Response

Reviewer #1     (R1)                                                                                                                  .

R1: L8 – delete „.“

A: Thank you very much for your correction. It has been done.

R1: Figure 1 is strange and should be adapted/simplified/splitted for the readers

A: Many thanks for your advice. We have simplified figure 1 following your suggestions, eliminating some elements. We hope that these changes will facilitate its visualization.

R1: Figure 3, Fig 4, Fig 5 – no references are included

A: Thanks for your comment. We are not sure if we have understood correctly. We have elaborated the three figures (Fig.3, Fig.4 and Fig.5) based on the information gathered during the review work. Then, we consider that we do not to include any references. Tables 1 and 2 show this information with the corresponding references. This information was added in the figure captions.

R1: Table 1 should be simplified, the clan-up part can be deleted

A: Thank you very much for your suggestion. We consider that the "Clean-up" is a relevant information in the analytical extraction procedures described. Therefore, we thought to delete the "Separation equipment" and "Mobile phases" columns to simplify the table. We have added the complete table (Table S1) to the supplementary material in order to ensure that this information is not lost. This was indicated in the main text. However, we are open to any further modification.

R1:  In Table2 samples can be divided under one name, not to repeat itself constantly. Volumes can be deleted.

A: Thank you very much for your comment. In the table, the techniques showed are ordered by type of samples and by article.  In some studies, two or three types of biological matrices were analyzed. For this reason, we considered to keep the column “Matrix” as presented in order to maintain the fluidity of the reading. In order to simplify the table, we have decided to delete the "Separation equipment" and "Mobile phases" columns as well.  We believe that this will ensure a homogeneous format for the two tables. We have also added the complete table (Table S2) to the supplementary material in order to ensure that this information is not lost. This will be indicated in the main text.

R1: The manuscript sounded interesting at first, but after closer examination, I found it difficult to follow. The references were not mentioned in figures. Tables are extensive and should be simplified. There is no need to repeat information in the text and in tables.

A: Thank you very much for your comments and suggestions. As we explained in a previous question, Figures 3, 4 and 5 have been elaborated with the information collected during the review task, shown in Tables 1 and 2. This will be noted in the figure captions. In addition, we have revised the text to summarize or omit information that is repeated or considered redundant. Changes will be noticed in the text using the "Track Changes". We really hope that these modifications, together with the reduction of the figures and tables, will simplify the follow-up of the manuscript.

R1: I would expect more systematical approach to the topic, without repeating. The time frame of the review should be mentioned in the title (from 2018). In general, the title should be more focused on what the manuscript is about.

A: Thank you very much for your comments. As a result of this revision process, changes have been made to the body of the manuscript, figures and tables. We really hope that the result of these modifications will provide a more structured study. The title has also been changed to " Non-occupational exposure to pesticides: experimental approaches and analytical techniques (from 2019)", in order to focus it more on the topic and the time frame.

R1: Supplementary material is hard to follow, especially tables S1 and S2 with all these formulas. Could this be simplified or removed?

A: Tables S1 and S2 has been removed and the main rational behind the calculations of the exposure has been explained in the text.

R1: In general, the manuscript is not bad, but it is confusing with potentially overloaded infos.

A: We appreciate the observation. As commented in previous questions, tables, figures and text have been revised and reduced in some aspects in order to minimize overload information.

Reviewer 2 Report

This manuscript is an educative summary of very recent studies on the analysis of pesticide residues for human exposure assessment. The comprehensive review will be of help for researchers to design experimental methods for pesticides and their transformation products to assess direct and indirect exposure. It seems that it is a bit inevitable to write a long article, but it can be formatted more concisely in some parts. Overall, the manuscript is acceptable as a review article after a few moderate revisions.

Formatting: Page numbers are labelled inappropriately and Table 2 should be formatted landscape as was Table 1.  

Line 112: It cannot be said “all studies published”. Time window from January 2019 to March 2021 is too narrow and may miss important results. Please justify. In line 627, it is written from “January 2018”. Please check the correct date.

Line 517: 39 protocols can be grouped. They are not all independent.

Section 5 can be moved right after introduction.

Author Response

Reviewer #2  (R2)                                                                                                                    .

R2: This manuscript is an educative summary of very recent studies on the analysis of pesticide residues for human exposure assessment. The comprehensive review will be of help for researchers to design experimental methods for pesticides and their transformation products to assess direct and indirect exposure. It seems that it is a bit inevitable to write a long article, but it can be formatted more concisely in some parts. Overall, the manuscript is acceptable as a review article after a few moderate revisions.

A: Thank you very much for your considerations. We really hope that this revision will facilitate the development of methodologies and analytical protocols to assess human exposure to pesticides from different perspectives. As a result of this review process, the text has been reorganized and some aspects have been reduced to make it easier to read.

R2: Formatting: Page numbers are labelled inappropriately and Table 2 should be formatted landscape as was Table 1.

A: Thanks for your comment. The orientation of Table 2 (vertical) was an error when integrating the manuscript into the journal template. We have changed its orientation so that both tables are displayed in landscape mode. This reduces the article's extension as well as being more visually attractive to the reader. In addition, we would like to notify that we were unable to change the page numbers within the template, so they continue to be mislabelled. In relation to this we request help/editing from the journal Molecules.

R2: Line 112: It cannot be said “all studies published”. Time window from January 2019 to March 2021 is too narrow and may miss important results. Please justify. In line 627, it is written from “January 2018”. Please check the correct date.

A: Thank you very much for your suggestion.  We have changed the words to “the most recent literature”, (line 114). Concerning the time frame in which the bibliographic search was carried out, we are sorry for the confusion. We conducted a primary search for articles published between January 2019 and March 2021, focusing on analytical studies of pesticides in environmental matrices and human exposure assessment. In addition, we then conducted a second search for previous reviews published in the same time frame, and a third and final search for specific wastewater-based epidemiology (WBE) studies focused on human exposure to pesticides. The latter was quite extensive, including articles published from January 2018 to March 2021. We have modified the section “Material and methods” in order to clarify this question.

R2: Line 517: 39 protocols can be grouped. They are not all independent.

A: Thanks for your suggestion. It has been done. It was reported (line 553.)

R2: Section 5 can be moved right after introduction.

A: Thank you very much for your suggestion. We have moved section 5 ("Materials and methods") just after the introduction as suggested. For this reason, the number of the remaining sections have also been modified and the text has been moved to readjust figures and tables.

Round 2

Reviewer 1 Report

no comments